# Developing and evaluating the impact of a small group communication programme in improving the entrepreneurial competence and economic self-efficacy of smallholder farmers with art skills

Yao Chen[1], Siti Sarah Maidin[2], Santas Tsegyu[3,4,5], Kabiru Adebowale Tiamiyu[6], Ijeoma Pauline Ogbonne[7], Danjuma Mathew Yare[8], Happiness Kodichinma Ogiri[7]*, Verlumun Celestine Gever[8]

1 English Department, School of Foreign Languages, Hangzhou City University, Hangzhou, China, 2 Faculty of Data Science and Information Technology (FDSIT), INTI International University, Nilai, Negeri Sembilan, Malaysia, 3 Department of Journalism and Broadcasting, Faculty of Communication and Media Studies Nasarawa State University Keffi, Keffi, Nigeria, 4 Research Fellow, Faculty of Social Sciences, Media and Communications, Keffi, Nigeria, 5 Department of Media Management, University of Religion and Denominations, Qom, Iran, 6 Department of Mass Communication, Federal University Lokoja, Lokoja, Nigeria, 7 Institute of African Studies, University of Nigeria, Nsukka, Nigeria, 8 Department of Mass Communication, University of Nigeria, Nsukka, Nigeria

* sohappy4myself@gmail.com

## Abstract

The researchers in Study 1 conducted interviews among experts and developed a small group communication programme to be delivered in 24 months. In Study 2, a quasi-experiment was conducted involving 540 smallholder farmers in Nigeria to test the impact of the developed programme. The result showed that smallholder farmers with art skills who received the small group communication programme reported a significant improvement in their entrepreneurial competence and economic self-efficacy compared to smallholder farmers who did not receive the programme. A follow-up assessment after two years revealed the steady effectiveness of the developed programme.

## Introduction

Smallholder farmers are those with ownership of small plots of land who also engage in farming on a small scale using labour from family members. A team of researchers [1] reveal that 80% of Nigerian farmers belong to this category and contribute significantly to addressing the food needs of the nation. Despite the contribution of smallholder farmers to the food security of many developing countries, they have remained largely poor [2]. Smallholder farmers acknowledge that their farming alone is not able to provide them with the income they need to meet their daily needs, so some of them learn skills to complement with income from farming. Two factors are responsible for this; first, smallholder farmers have a limited plot of land to cultivate. In the second place, they use only labour from family members. In many

**Data Availability Statement:** The data for this study is available at https://www.openicpsr.org/openicpsr/project/194361/version/V1/view.

**Funding:** The authors received no specific funding for this work.

**Competing interests:** The authors have declared that no competing interests exist.

communities, smallholder farmers acquire art skills with the expectation that it will assist them to augment their income and improve their socio-economic ratings.

Although skill acquisition has been regarded in literature [3–5] as an important requirement for economic empowerment, if such skills are not marketed, they will not serve the interest of those who possess them. This situation is likened to a production process where it is said that production is not complete until it gets to the final consumer. A team of researchers [6] affirm that skill utilization is one of the surest ways that enable those with such skills to benefit from the skills they possess. Studies from different countries such as Brazil [7] China [8] Malaysia, [9] and Mexico [10] have all pointed to the fact that skill acquisition is essential for economic empowerment of people.

Within the context of this study, it can be said that when a person possesses a skill set, it is expected that such skills will assist him or her in the area of income growth. This is because skills are essential assets and those who possess them expect to benefit from such. A researcher [10] affirms that there is a significant relationship between skill possession and economic growth. A Another study [8] holds a similar view regarding the usefulness of skill acquisition in promoting economic growth.

Researchers [11, 12] pay greater attention to investigating issues related to technical skill acquisition as though skill acquisition by itself will automatically result in an improvement in income and the earning capacities of those with such skills. According to a study [13] entrepreneurial competence is essential for people to survive as well as market their potential with a view to improving their economic abilities. Another scholar [14] in making a case for the centrality of entrepreneurial competence says that one of the reasons businesses in Nigeria find it difficult to survive is because business owners do not have the required skills to operate effectively in a competitive environment. Another set of scholars [15] in making a case for the centrality of entrepreneurial competence aver that such skills will make people self-dependent and economically viable. Global Entrepreneurial Monitor (2012) cited in a study [14] says that when it comes to business spirit, Nigeria ranks high, but instances of failed business are equally high in Nigeria. It attributes such instances to low entrepreneurial competence. Small group communication is one of the approaches that can be used to improve the entrepreneurial competence and economic self-efficacy of smallholder farmers with art skills. This is because small group communication allows people to interact and exchange ideas in a close range. Small group communication enables people to set goals, pursue such goals, outline how to achieve the goals and evaluate their performance. A researcher [16] affirms that small group communication is an effective strategy that promotes interaction and knowledge internalization. From the submission of the study, it can be argued that small group communication can serve as a tool for guiding smallholder farmers to convert their skills into money. Therefore, the objective of this study was to develop and evaluate the impact of small group communication programme on the economic self-efficacy of smallholder farmers with art skills in Nigeria.

## Study objective and significance

The basic objective of this study was to develop and evaluate the impact of small group communication programme on the economic self-efficacy of smallholder farmers with art skills in Nigeria. A study of this nature is important to literature because previous studies have paid more attention to the importance of skill acquisition with less attention to how acquired skills can be marketed. Therefore, the current study has addressed this gap. Furthermore, this study has practical relevance because policymakers will find it useful as a guide for promoting the marketing of skills. This is important because if possessed skills are not monetized, they may not be of economic value to those who possess them.

## Effect of small group communication

Communication is central to business promotion. Without communication, it will be difficult for any meaningful business engagement to take place. Communication is at the heart of all marketing promotions. These are advertising, sales promotion, personal selling, publicity, and direct marketing respectively. It will be difficult, if not impossible, for any of the marketing promotions to take place without the use of communication. According to Scott and Walker [17] promotional activities have proven to be essential in ensuring the success of department stores in fending off competition from the expanding chain stores by drawing in customers to their large, central premises. The point to make here, therefore, is that communication is at the centre of business activities. Communication can take place under different circumstances involving different people. For example, when communication involves two persons on a face-to-face basis, it is called interpersonal communication [18] When it involves the use of mass media to a scattered, large, anonymous and heterogeneous audience, it is called mass communication [19]. Communication can also involve three or more persons who engage in a communication exercise to achieve a particular objective. Such communication is called small group communication.

One of the pioneers of small group communication is Bormann and Bormann [20]. According to Bormann and Bormann, small group communication describes the exchange of meaning involving two or three persons who have a particular objective to attain within a certain time frame. From the views of Bormann and Bormann, small group communication has two important elements. The first is that small group communication has a goal that participants wish to achieve. The second element is that such goals will be achieved within a certain time frame. The implication here is that small group communication has the potential to assist people to achieve goals. David and Chris [21] say small group communication occurs on a face-to-face or any other channel of communication in accordance with the channel of communication agreed on among participants. According to Fujishin [22], small group communication starts by first determining those who are involved in the communication engagement. When a membership is determined, it gives way for interaction among group members, which typically occurs in two dimensions. The first dimension is the task dimension which focuses on the events that will take place in the communication. Within the context of the task dimension, group members engage in some activities with the objective of ensuring that the goals of the group are achieved. The second dimension is the social dimension which describes the social relations that occur among group members in the course of the communication. According to Burtis and Turman [23], small groups mostly develop and utilize patterns aimed at providing direction for their communication engagement which Burtis and Turman [23] called communication networks. The fundamental thing about small group communication is that it is driven by goals. Within the context of this study, the goal of using small group communication will be to improve entrepreneurial competence and economic self-efficacy.

Small group communication has the potential to serve as an effective tool for behaviour change. A team of researchers [11] conducted a study to assess the impact of small group communication in influencing behaviour intention regarding art skills. The researchers utilized a quasi-experiment design to achieve their objectives. They examined 470 respondents who were victims of conflict. The art skill that was used in the study was painting while the researchers applied a *t*-test to analyse the results of the study. They found that small group communication was an effective communication tool for promoting skill acquisition among victims of conflict. Another team of researcher [24] also reported that small group was effective in propelling behaviour intention towards business start up.

Madonsela [25] carried out a study in which they tested the impact of small group communication in assisting formal organizations to achieve their goals. The researchers utilized a survey research design and reported that small group communication is capable of aiding organizations to achieve their objectives. The result of the study suggests that small group communication is an effective behaviour change tool. Lowry et al., [26] conducted a study wherein they ascertained the effect of different group sizes as well as social presence on small-group communication. The design that was used for the study was a quasi-experiment with a questionnaire as the instrument for data collection. The result of the study showed that small group communication is an effective behaviour change tool. Another study worth mentioning here is that of Lamport and Rynsburger [27] who reported that small group communication is an efficient tool for evangelism. The critical point to note from the above-reviewed studies is that small-group communication has the potential to improve the entrepreneurial competence and economic self-efficacy of smallholder farmers with art skills.

## Art skills among farmers, entrepreneurial competence, and economic self-efficacy

In most traditional African societies, smallholder farmers also possess art skills to enable them to meet their needs for some items. Art skills describe the ability to create work of art that will be of benefit to mankind. Examples of some art skills include drawing, weaving, musical composition, sculpting and painting. Smallholder farmers who possess art skills combine it with farming with the expectation that this will assist them in living a better life. However, their low knowledge of entrepreneurial principles makes it difficult for them to maximize the potential inherent in the skills they possess. Troncoso and Vergara [28] reveal that from time immemorial, farmers have engaged in art-making as a deliberate strategy to diversify their income and improve their standard of living. Kei and Line [29] affirm that many rural farmers acknowledge the importance of art in their lives.

Entrepreneurship has been regarded as critical to the; economic development of any nation. A team of researchers [30] affirm that entrepreneurship is the life-wire of economic development in any nation. Conceptually, entrepreneurship entails starting and managing a business entity. According to Barot [31], entrepreneurship is important to everybody and anyone who begins a new business entity is venturing into the world of entrepreneurship. According to Chang and Wyszomirski [32], art entrepreneurship is gradually becoming common and it entails converting knowledge of artwork into business ventures.

Suffice it to say that entrepreneurship entails engagement in business activities, then it follows logically that entrepreneurial competence is important for successful entrepreneurial activities. Such skills are needed so that entrepreneurs will be successful at what they do. Evidence in the literature [33, 34] points to the fact that entrepreneurship competence is an essential requirement for successful entrepreneurial activities. A team of researchers [34] say that entrepreneurial competence, not certificate, determines the success of businesses.

When entrepreneurs develop competence, it is expected that such skills will positively impact economic self-efficacy. That is to say that it is expected that the possession of entrepreneurial competence will translate to success in business and eventually income. Economic self-efficacy can be defined as the ability of a person to meet their financial needs independently of other people. Hoge et al. [35] note that economic self-efficacy is important because it allows people to meet their daily needs while also maintaining a sense of self-worth. Elise and Ellen [36] opine that economic self-efficacy plays a significant role in influencing the type of relationship that exists among people. The central point to note here is that the economic self-efficacy of smallholder farmers is crucial in determining their standard of living, hence the need to suggest ways of improving their economic self-efficacy.

## Theoretical framework

The researchers applied the Human Capital Theory (HCT) to conduct the current study. Gary Stanley Backer propounded the human capital theory in 1964 to explain the importance of skill acquisition in empowering people [37]. The basic assumption of the human capital theory is that skill sets are just as important as money such that when people acquire saleable skills, it helps them to improve their economic fortunes. The theory assumes that one of the ways to promote economic development is by equipping people with skills that will make them self-dependent and sustaining.

Generally, researchers [5, 38–40] agree that skill acquisition is an important strategy for empowering people. Obasi et al. [41] applied the human capital theory to ascertain the impact of theatre for development in improving the acquisition of skills in painting and weaving among a sample of 470 victims of conflict. The result of the study showed that the theory was an appropriate framework for assessing issues related to skill acquisition. Okpara et al. [12] equally applied the story to examine the acquisition of skills among jobless youth in Nigeria and reported that the framework provides a sufficient basis for understanding the importance of skill acquisition and how people can acquire skill sets. The argument that the human capital theory advances suits very well with the current study because when people are equipped with entrepreneurial competence, it will have a corresponding positive effect on their economic self-efficacy. When people acquire skills without using them, the essence of human capital development is defeated. The theory, thus, assumes that people who are equipped with skills will use such skills to make life better for themselves. Smallholder farmers with art skills need to make use of such skills so that they can actually be empowered. The development of human capital requires the use of appropriate tools. Within the context of this study, small group communication is regarded as a useful tool for developing the human capital of smallholder farmers. According to Gever et al. [11], small group communication is an effective tool for assisting individuals to develop skills aimed at making them economically useful to themselves and their immediate environment. Arias et al. [42] conducted a study wherein they compared small group communication with a lecture format and reported that the former was a far more effective strategy for promoting skill acquisition than the latter.

Based on this theory, the researchers hypothesized:

**H1:** Smallholder farmers who take part in a small group communication intervention will score higher means regarding their entrepreneurial competence than those who do not.

**H2:** Smallholder farmers who take part in a small group communication intervention will score higher means regarding their economic self-efficacy than those who do not.

## Methodology

### Study 1

In the first study, the researchers sought to develop a small group communication programme that could be used to improve the entrepreneurial competence and economic self-efficacy of smallholder farmers with art skills. Therefore, content analysis was the design of the study. The participants (n = 33) were experts in agricultural economics (n = 8), mass communication (n = 7), counselling (n = 7), social work (n = 5), and entrepreneurship (n = 6). The researchers recruited the participants using the respondents-driven chain referral sampling technique. This was done by first recruiting earlier participants who were requested to recommend potential participants. The instrument of data collection was a face-to-face interview that lasted between 30 minutes to one hour. Each interview began with a broad question- How can small

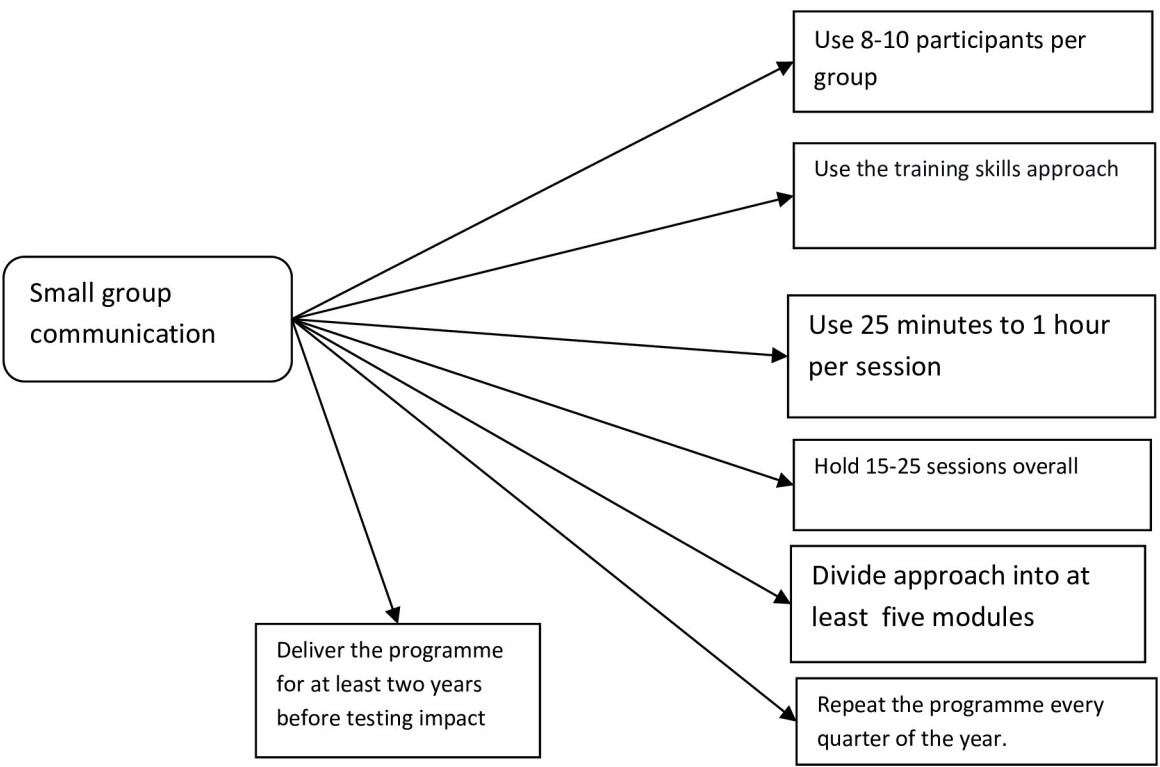

**Fig 1. A programme for using small group communication to improve entrepreneurial competence and economic self-efficacy of smallholder farmers with art skills.**

group communication be used to improve entrepreneurial competence and economic self-efficacy of smallholder farmers with art skills? Follow-up questions were asked to guide the interview session. Based on the interview session, Fig 1 reveals a developed programme for improving entrepreneurial competence and economic self-efficacy of smallholder farmers with art skills.

In Fig 1 above, the researchers present a small group communication programme that could be used to improve the entrepreneurial competence and economic self-efficacy of smallholder farmers. It is important to clarify here that the training skills take the education approach where the focus is to educate the participants on an issue.

## Study 2

In study 2, the researchers attempted to test the effectiveness of the developed programme. The design that was used in this study was a quasi-experiment. A quasi-experimental design is mostly required when a non-laboratory experiment is needed to test the effectiveness of an intervention [43–48]. The participants in this study were 540 smallholder farmers with art skills. Before deciding on the sample size, the researcher carried out a priori power analysis using the G*power programme. The output from this analysis showed that the researchers needed a sample size of 540 to determine the existence of a relationship at 0.05 level of significance. The study was carried out in Ede-Oballa, in Nsukka Local government area of Enugu State, Nigeria. The area has two autonomous communities namely: Ede-Ukwu and Ede-Enu. Combined, both communities have fifty-two villages, twenty-two from Ede-Enu and thirty from Ede-Ukwu. The study announcements were made at the village halls and those interested

were asked to fill forms and submit. The forms sought information on the art skills and farming activities, The information submitted was evaluated by a team of 10 research administrators. The team was made up of two experts each in measurement and evaluation, fine and applied arts, communication, agricultural economics and business education. The mandate of the research administrators was to ensure that the right participants were included in the sample. Initially, 1500 indicated interest to participate, but during screening, only 1,080 participants were eligible. The researchers needed only 540 participants so systematic sampling technique was used. This was done using the following systematic approach:

1. The smallholder farmers were numbered from 1–1080.

2. The sample size, n was decided to be 540.

3. The interval, k = 2. That is $2 \times 540 = 1,080$ and $\frac{1080}{540} = 2$.

Every second participant was selected and this continued until all the 540 participants were sampled. The participants were then randomly assigned to control and treatment groups made up of 270 participants each.

The researchers used the entrepreneurial competence scale and economic self-efficacy scale to collect data for the study.

**Entrepreneurial competence scale.** This scale was developed by Silveyra et al. [49] to assess economic competence. It has nine items spread across three dimensions namely competence in identifying opportunities (3 items; $\alpha = 0.88$), evaluation of identified opportunities (3 items; $\alpha = 0.89$), and exploitation of opportunities (3 items; $\alpha = 84$). The overall reliability was .87. Also, a pilot study with 30 participants showed an overall Cronbach's alpha of .77, meaning that it was okay for the Nigerian sample.

**Economic self-efficacy scale.** Hoge et al. [35]) developed this scale with 10 items to assess the financial independence of individuals. Hoge et al. reported a Cronbach's reliability coefficient of .88. The researchers also conducted a pilot study in Nigeria with 30 participants and this revealed a Cronbach's reliability coefficient of .86 which was regarded as acceptable.

## Procedure for the experiment

To conduct this experiment, participants in the treatment group took part in a small group communication intervention. There were 27 small groups made up of 10 members each. Each group had a moderator who was an expert in entrepreneurship. The discussion focused on developing the entrepreneurial competence of respondents. The respondents organized the discussion in six modules that were organized based on entrepreneurial competence. In module one, the attention was on the importance of entrepreneurship. In the second module, the discussion focused on farming and art skills as entrepreneurial ventures. In the third module, attention was paid to the appropriate entrepreneurial competences. The fourth module was on how to become successful entrepreneurs while the fifth module was on challenges of entrepreneurship. Module six focused on how to overcome challenges associated with entrepreneurship. Each of the modules was delivered three times, making it a total number of 18 sessions each year and 36 sessions for the two years. The researchers collected data for the study three times. The first was before the intervention, the second was after the intervention, and the last time was a year follow-up assessment.

## Data analysis

To conduct the analysis in this study, the researchers utilized Statistical Package for Social Sciences version 22. Furthermore, the researchers deployed simple percentages, mean and

standard deviation as well as a two-way analysis of variance (ANOVA) with repeated measures to explore the impact of the programme at pre-stage and post-treatment.

### Ethical approval

Ethical approval was sought and received from Ianna Research Academy. The type of approval was exempt because participation in the study did not expose the responsdents to danger.

### Results

The return rate for this study was 94% for the control group and 92% for the treatment group. All the respondents were smallholder farmers with art skills. This means that the respondents had art skills that they could combine it with farming to make their lives better. The mean age of the respondents was 33 years. The result of Table 1 revealed the impact of the programme on the entrepreneurial competence and economic self-efficacy of smallholder farmers with art skills.

Table 1 was computed to ascertain the entrepreneurial competence and economic self-efficacy of smallholder farmers who took part in the study. The result of the study showed low entrepreneurial competence and economic self-efficacy among the participants with no significant difference detected between the treatment and control group at pre-treatment. After the programme, participants in the treatment group reported a significant improvement in their entrepreneurial competence and economic self-efficacy, but those in the control group did not report a significant improvement even after a follow-up assessment after two years. Based on this result, hypotheses 1 and 2 were supported. The graphical illustration of the result is further presented in Figs 2 and 3 below:

### Discussion of findings

In this study, the researchers set out to develop and measure the effectiveness of a small group communication programme in improving the entrepreneurial competence and economic self-efficacy of smallholder farmers who possess art skills. Two independent but complementary studies were conducted. The first study was an exploratory effort to develop a small group communication programme for improving the entrepreneurial competence of smallholder

**Table 1. Pre-intervention, post-intervention and follow-up ANOVA results on entrepreneurial competence and economic self-efficacy of smallholder farmers with art skills.**

| | Entrepreneurial competence | | | | | Economic self-efficacy | | | | |
|---|---|---|---|---|---|---|---|---|---|---|
| | *Mean* | *SD* | *P* | *t* | *Dec.* | Mean | SD | *P* | *T* | Dec |
| **Pre-intervention** | | | | | | | | | | |
| Control group | 16.3 | 3.8 | | | | 10.6 | 1.5 | | | |
| Treatment group | 16.1 | 3.8 | | | | 10.7 | 2.4 | | | |
| | | | .707 | 375 | Not sig | | | .379 | .881 | Not sig |
| **Post-intervention** | *Mean* | *SD* | *P* | *t* | *Dec.* | Mean | SD | *P* | *T* | Dec |
| Control group | 16.2 | 3.6 | | | | 10.7 | 1.5 | | | |
| Treatment group | 36.8 | 5.4 | | | | 27.5 | 9.5 | | | |
| | | | .001 | 125.535 | Sig | | | 0.001 | 29.241 | Sig |
| **Follow-Up assessment** | *Mean* | *SD* | *P* | *t* | *Dec.* | Mean | SD | *P* | *T* | Dec |
| Control group | 17.3 | 3.8 | | | | 11.1 | 2.5 | | | |
| Treatment group | 38.5 | 5.9 | | | | 30.8 | 6.1 | | | |
| Total average | | | .001 | 120.837 | Sig | | | 0.001 | 103.681 | Sig |

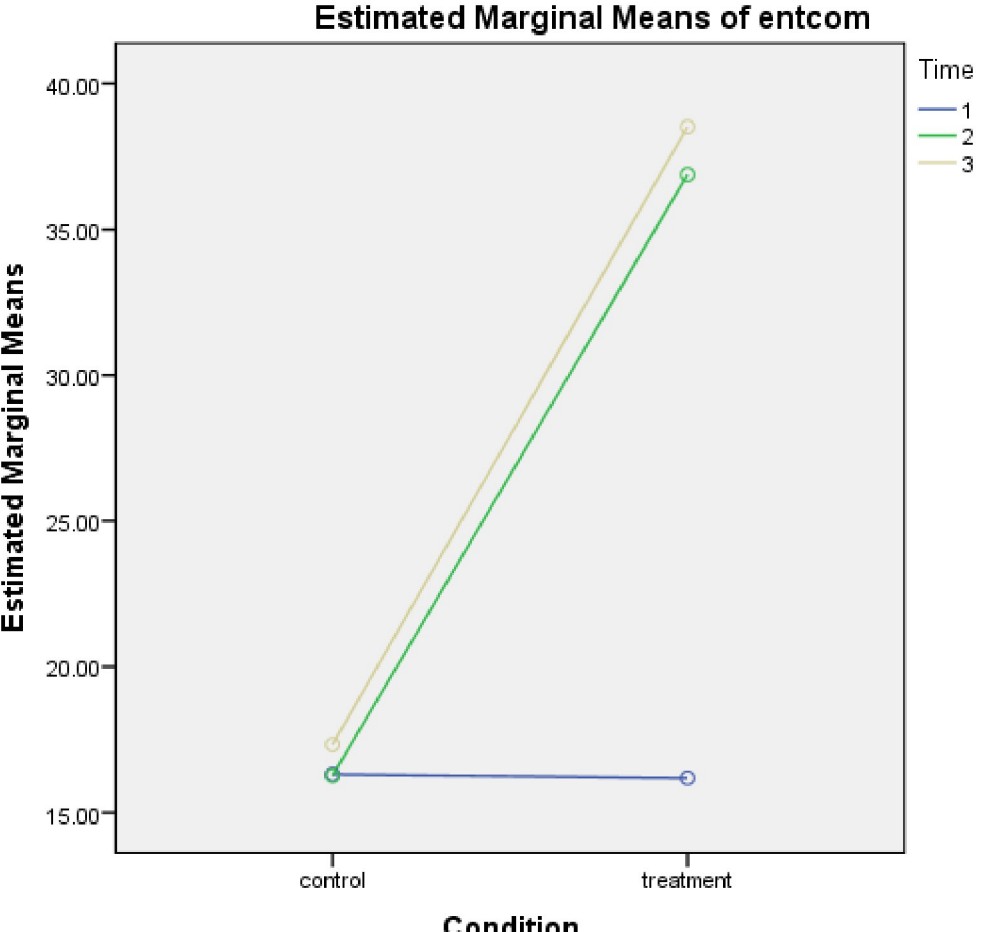

**Fig 2. Entrepreneurial competence of the participants at before, after and follow-up assessment.**

farmers with art skills. After the programme was developed, its effectiveness was tested on a sample of 540 smallholder farmers with art skills in the areas of weaving, sculpture and music. The result of the study showed that before the treatment was administered to the respondents, both the treatment and the control groups scored low regarding entrepreneurial competence and economic self-efficacy. What this means is that even though smallholder farmers acquire art skills, they lack entrepreneurial competence and are unable to improve their economic self-efficacy. This highlights the need to assist them so that they will be able to increase their earnings. It is not sufficient to possess skills; such skills must be able to put food on the table for those who possess them. Okpara et al. [12] in a study reported that many people with art skills do not have jobs. Accordingly, they argued that entrepreneurship, especially in arts, is one of the ways that people with art skills can improve their income. Gever et al. [11] in their study also reported that art skills, when properly applied, can be useful in economic empowerment.

Furthermore, the result of this study showed that after the intervention, there was a significant mean difference between the control and the treatment groups. In particular, it was found that respondents in the treatment groups scored higher regarding their entrepreneurial competence and economic self-efficacy. What this means is that small group communication played a useful role in assisting participants to improve their skills as well as income. This result has extended studies [25–27] related to the effectiveness of small group communication

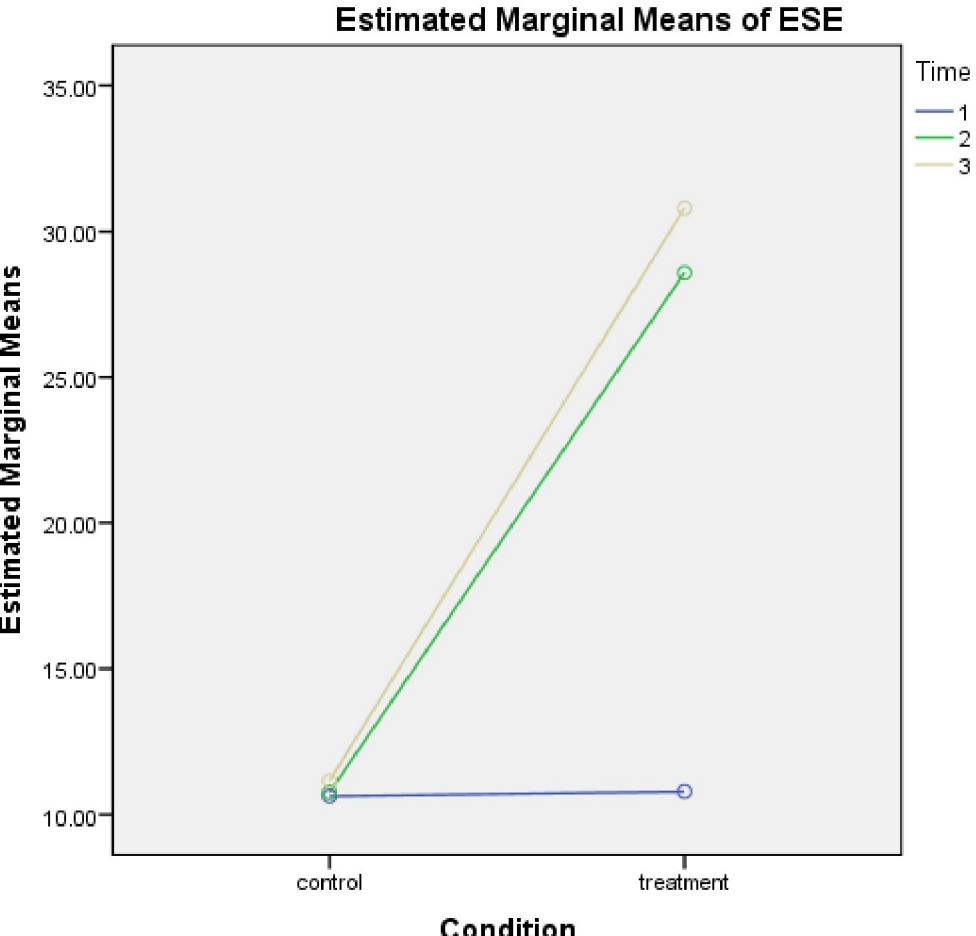

**Fig 3. Economic self-efficacy of the participants before, after and follow-up assessment.**

by adding how it can play a critical role in entrepreneurship and income. This is because most of the previous studies on the effectiveness of small group communication pay less attention to issues related to income. Additionally, the result of the current study has extended previous studies [12, 49] related to skill acquisition by pointing out the need to convert acquired skills into income. It is one thing to acquire skills and another to ensure that such skills are useful for those who acquire them.

## Implications of the results

The result of this study has three broad implications namely: scholarship, theory and practice. Scholarly, this study has extended the argument regarding the importance of small group communication as an intervention tool, the need for entrepreneurial competence as well as how this is linked to economic self-efficacy. This new dimension which links small group communication with income and entrepreneurial competence will shape future debates in the literature on communication effectiveness, business, and art skills.

The result of this study also has implications for the connection between art skills and entrepreneurial ability. Art skills are good but they may not be able to serve those who possess them if they do not have the corresponding entrepreneurial competence to market those skills.

Skills are assets and can only be useful to those who possess them if they are able to convert them into money. Therefore, this study has shown that there is a need to train those with technical skills in the area of entrepreneurship so that they will benefit from the skills that they have.

Furthermore, the result of this study has implications for the human capital theory by showing that human capital development should not be limited to skills acquisition but also extended to the utilization of such skills to make money. This is important because skills on their own do not translate to anything if they are not maximized to the benefit of those who possess them. This aspect of the current study has added a new perspective to the human capital theory by showing that programmes are needed to move from skill acquisition to skill utilization.

The result of the current study has practical implications by highlighting the need for change agents in both government and non-profit organizations to utilize small group communication as a behaviour change tool. This is because small group communication allows for close interaction among people. It offers people the opportunity to ask questions and receive answers. This is not possible in mass media communication.

Overall, this study has filled a gap in the literature by going beyond examining the importance of skill acquisition by providing empirical evidence that could be useful in developing learning packages for assisting smallholder farmers to improve their economic self-efficacy.

## Conclusion/ suggestions for further studies

Based on the result of this study, the researchers conclude that small group communication can serve as a useful tool for assisting smallholder farmers to acquire entrepreneurial competence and improve their income. The basic contribution of this study is that it has provided empirical evidence that not only explains the impact of small group communication in improving business performance but has highlighted the need to encourage smallholder farmers with art skills to convert their skills into money. Despite this, the current study has some limitations. First, the researchers examined only smallholder farmers, it will be beneficial to examine other categories like university graduates with certification in vocational and technical skills. The researchers did not also examine other skill sets apart from arts. Further studies should look at these limitations. It is also suggested that further studies should apply social exchange theory to examine the impact of small group communication on behaviour change.

## Author Contributions

**Conceptualization:** Kabiru Adebowale Tiamiyu, Verlumun Celestine Gever.

**Data curation:** Happiness Kodichinma Ogiri, Verlumun Celestine Gever.

**Project administration:** Siti Sarah Maidin.

**Resources:** Yao Chen, Siti Sarah Maidin.

**Software:** Kabiru Adebowale Tiamiyu.

**Supervision:** Santas Tsegyu.

**Validation:** Santas Tsegyu.

**Writing – original draft:** Ijeoma Pauline Ogbonne, Danjuma Mathew Yare.

**Writing – review & editing:** Ijeoma Pauline Ogbonne, Happiness Kodichinma Ogiri.

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
