## [Decision Letter · Decision Letter 0]

26 Jul 2023

PONE-D-23-15217Developing and evaluating the impact of a small group communication programme in improving the entrepreneurial competence and economic self-efficacy of smallholder farmers with art skillsPLOS ONE

Dear Dr. Gever,

Thank you for submitting your manuscript to PLOS ONE. After careful consideration, we feel that it has merit but does not fully meet PLOS ONE’s publication criteria as it currently stands. Therefore, we invite you to submit a revised version of the manuscript that addresses the points raised during the review process.

Authors to please proof read the paper very carefully. 

We look forward to receiving your revised manuscript.

Kind regards,

Muhammad Khalid Bashir, PhD

Academic Editor

PLOS ONE

Journal Requirements:

"The study was not funded."

Please include your amended statements within your cover letter; we will change the online submission form on your behalf."

"We do not have a competing interest."

5. Please ensure that you include a title page within your main document. You should list all authors and all affiliations as per our author instructions and clearly indicate the corresponding author.

Reviewers' comments:

Reviewer's Responses to Questions

**Comments to the Author**

1. Is the manuscript technically sound, and do the data support the conclusions?

Reviewer #1: Yes

Reviewer #2: Yes

2. Has the statistical analysis been performed appropriately and rigorously? 

Reviewer #1: Yes

Reviewer #2: Yes

3. Have the authors made all data underlying the findings in their manuscript fully available?

Reviewer #1: Yes

Reviewer #2: Yes

4. Is the manuscript presented in an intelligible fashion and written in standard English?

Reviewer #1: Yes

Reviewer #2: Yes

5. Review Comments to the Author

Reviewer #1: The paper suggests promising interventions to effectively improving entrepreneurial capacities and economic self efficacy of smallholder farmers. The writing style is clear and concise; however, some minor grammatical and spelling errors are present. The introduction provides a coherent overview of the objectives and scope of the paper, well-structured and provides a comprehensive examination of the small communication programme. The conclusion summarizes the results and recommendations of the paper succinctly.

Reviewer #2: Dear Author,

First of all I would like to say that the study presents the results of original research. The Experimental study is important for such kinds of topics and the author was successfully targeted the right sample with proper methodology. However, the following comments are important to be considered to further improve the current study:

1- Specify what art skills does the author refers, the author has to specify it.

2- The literature should better explain measurements of self-efficiency, entrepreneurial competencies and dimentions:

Read:

i. Developing and Validating the Scale of Economic Self-Efficacy

ii. Effects of Intellectual Capital on the Performance of Malaysian Food and Beverage Small and Medium-Sized Enterprises

iii. Developing self-efficacy and career optimism through participation in communities of practice within Australian creative industries

iv. Developing and testing a social media-based intervention for improving business skills and income levels of young smallholder farmers

3- The author needs to develop the conclusion and provides more details on his findings and contribution.

4- It would be much better of the author provides contradicted literature and justify his findings in discussion section.

6. PLOS authors have the option to publish the peer review history of their article (what does this mean?). If published, this will include your full peer review and any attached files.

Reviewer #1: **Yes: **Saima Afzal, PhD

Reviewer #2: **Yes: **Nabaz Nawzad Abdullah

---

## [Author Response · Author response to Decision Letter 0]

1 Sep 2023

5. Review Comments to the Author

Comment: Reviewer #1: The paper suggests promising interventions to effectively improving entrepreneurial capacities and economic self efficacy of smallholder farmers. The writing style is clear and concise; however, some minor grammatical and spelling errors are present. The introduction provides a coherent overview of the objectives and scope of the paper, well-structured and provides a comprehensive examination of the small communication programme. The conclusion summarizes the results and recommendations of the paper succinctly.

Response: The manuscript has been proof-read thoroughly for grammar.

Reviewer #2: Dear Author,

Comment: First of all I would like to say that the study presents the results of original research. The Experimental study is important for such kinds of topics and the author was successfully targeted the right sample with proper methodology. However, the following comments are important to be considered to further 

improve the current study:

Comment: 1- Specify what art skills does the author refers, the author has to specify it.

Response: This has been done and specified.

Comment: 2- The literature should better explain measurements of self-efficiency, entrepreneurial competencies and dimentions:

Read:

Response: We have done this and highlighted in red. 

i. Developing and Validating the Scale of Economic Self-Efficacy

Response: This has already been cited in the document. See methodology and reference list. 

Effects of Intellectual Capital on the Performance of Malaysian Food and Beverage Small and Medium-Sized Enterprises. 

Response: This has been cited now in both the document and the reference list. 

iii. Developing self-efficacy and career optimism through participation in communities of practice within Australian creative industries

Response: This has been cited in the reference and the document.

iv. Developing and testing a social media-based intervention for improving business skills and income levels of young smallholder farmers

Response: This has been cited in the document and reference list. 

Comment: 3- The author needs to develop the conclusion and provides more details on his findings and contribution.

Respond: We have included a conclusion and the contribution of the study. This has been highlighted in red. 

Comment: 4- It would be much better of the author provides contradicted literature and justify his findings in discussion section.

Response: This has been done. We have shown how the current study differs with the previous ones. This has been highlighted in red.

---

## [Decision Letter · Decision Letter 1]

26 Sep 2023

Developing and evaluating the impact of a small group communication programme in improving the entrepreneurial competence and economic self-efficacy of smallholder farmers with art skills

PONE-D-23-15217R1

Dear Dr. Ogiri,

We’re pleased to inform you that your manuscript has been judged scientifically suitable for publication and will be formally accepted for publication once it meets all outstanding technical requirements.

Kind regards,

Muhammad Khalid Bashir, PhD

Academic Editor

PLOS ONE

Additional Editor Comments (optional):

Reviewers' comments:

Reviewer's Responses to Questions

**Comments to the Author**

1. If the authors have adequately addressed your comments raised in a previous round of review and you feel that this manuscript is now acceptable for publication, you may indicate that here to bypass the “Comments to the Author” section, enter your conflict of interest statement in the “Confidential to Editor” section, and submit your "Accept" recommendation.

Reviewer #1: All comments have been addressed

2. Is the manuscript technically sound, and do the data support the conclusions?

Reviewer #1: Yes

3. Has the statistical analysis been performed appropriately and rigorously? 

Reviewer #1: Yes

4. Have the authors made all data underlying the findings in their manuscript fully available?

Reviewer #1: Yes

5. Is the manuscript presented in an intelligible fashion and written in standard English?

Reviewer #1: Yes

6. Review Comments to the Author

Reviewer #1: (No Response)

7. PLOS authors have the option to publish the peer review history of their article (what does this mean?). If published, this will include your full peer review and any attached files.

Reviewer #1: **Yes: **Dr. Saima Afzal

---

## [Editor Report · Acceptance letter]

24 Oct 2023

PONE-D-23-15217R1 

Developing and evaluating the impact of a small group communication programme in improving the entrepreneurial competence and economic self-efficacy of smallholder farmers with art skills 

Dear Dr. Ogiri:

I'm pleased to inform you that your manuscript has been deemed suitable for publication in PLOS ONE. Congratulations! Your manuscript is now with our production department. 

Kind regards, 

on behalf of

Dr. Muhammad Khalid Bashir 

Academic Editor

PLOS ONE